# Hybrid Random Forest Survival Model to Predict Customer Membership Dropout

Pedro Sobreiro [1,2,*,†] , José Garcia-Alonso [2] , Domingos Martinho [3] and Javier Berrocal [2,†]

1   Sport Sciences School of Rio Maior (ESDRM), Polytechnic Institute of Santarém, 2001-904 Santarém, Portugal
2   Quercus Software Engineering Group, University of Extremadura, 06006 Badajoz, Spain
3   ISLA Santarém, 2000-241 Santarém, Portugal
*   Correspondence: sobreiro@esdrm.ipsantarem.pt; Tel.: +351-935585561
†   These authors contributed equally to this work.

**Abstract:** Dropout prediction is a problem that must be addressed in various organizations, as retaining customers is generally more profitable than attracting them. Existing approaches address the problem considering a dependent variable representing dropout or non-dropout, without considering the dynamic perspetive that the dropout risk changes over time. To solve this problem, we explore the use of random survival forests combined with clusters, in order to evaluate whether the prediction performance improves. The model performance was determined using the concordance probability, Brier Score and the error in the prediction considering 5200 customers of a Health Club. Our results show that the prediction performance in the survival models increased substantially in the models using clusters rather than that without clusters, with a statistically significant difference between the models. The model using a hybrid approach improved the accuracy of the survival model, providing support to develop countermeasures considering the period in which dropout is likely to occur.

**Keywords:** customer dropout; machine learning; survival analysis

## 1. Introduction

Customer retention is a problem that many organizations have to deal with, in the context of which dropout prediction provides insights to identify customers that could churn. Dropout represents the decision of a customer to end their relationship with an organization [1], which creates two outcomes: Dropout or non-dropout. The case where dropout is developed has two main scenarios [2,3]: (1) Contractual settings, where customers pay a monthly fee and the customer informs the end of the relationship; and (2) non-contractual settings, where the organization has to extrapolate whether the customer is still active or not. In the contractual setting, the customer must choose whether they will dropout or not; for example, if they renew a contract or not [4]. This means that, in contractual settings, the customer dropout represents an explicit ending of a relationship that is more penalizing than that in non-contractual settings [5], which has implications for the profitability of organizations, increasing marketing costs and reducing sales [6].

The advantages of developing retention strategies have been supported in the concept that the costs of customer retention are lower than those associated with customer acquisition [7,8], where a reduction of dropout by 5% could realize almost a duplication of profits [9]. To address this problem, the use of the customer databases could be explored, which is considered the most valuable asset that most organizations possess [10]. The development of a customer retention strategy could be supported through the identification of customers may dropout [11]; for example, using churn prediction models to detect customers with high propensity to dropout [12].

The anticipation of the dropout allows for the development of countermeasures to reduce customer churn. Several studies have addressed the problem related to customer retention in trying to improve the profitability [13–15]; in particular, organizations have

been addressing this problem by shifting their target from capturing new customers to preserving existing ones [14], considering that investments in retention strategies are more profitable than acquiring new customers [13].

Machine learning allows for the extraction of patterns from data, learning from a model using a set of descriptive features and a target feature based on a set of historical examples [16]. The approaches normally employed address the problem by use of a dependent variable representing dropout or non-dropout, without considering the dynamic perspective that the dropout risk changes over time [11]. A static perspective determines the dropout risk at a specific moment in time, but does not consider changes over time. Survival models have been proposed in an attempt to resolve this limitation [17], capturing the temporal dimension of customer dropout to predict when the dropout will occur, as well as the use of censured data, which allows for consideration of existing information about customers that have not churned yet [18].

There are several challenges related to the timing of the dropout, such as considering the behavior of the customer as static and not considering the dynamic behavior of the customer, in terms of the intent to dropout [11]. The importance of understanding when dropout will occur and the risk related to the temporal perspective of the problem seems to be an element that should be addressed. However, few studies have considered this aspect [18–21]. Van den Poel and Larivière [20] have used a Cox proportional hazards model to investigate customer attrition in an European financial services company. Baesens et al. [21] have explored the use of a neural network-based survival model to anticipate the timing when customers will default or pay off loans early. Burez and Vandenpoel [19] have proposed two different processes to predict customer dropout, instead of using only one, including commercial and financial churn, and suggested that financial churn (customer stops paying the invoices) is easier to predict than commercial churn (customers not renewing their subscription). Perianez et al. [18] have investigated the use of survival trees and forests to improve prediction accuracy compared to traditional methods, such Cox regression, in mobile social game subscriptions.

Survival analysis, which originates from biomedical statistics, is especially well-suited to studying the timing of events in longitudinal data [22]. Survival analysis consists of a class of statistical methods modelling the occurrence and timing of an event, such as the customer dropout. Survival analysis allows us to examine not only if an event occurred, but also how long it took to occur. The primary value of survival analysis in our context, however, is comparing the dropout probability for individuals classified through theoretically relevant variables. The survival methods have enjoyed increasing popularity in several disciplines, ranging from medicine to economics [22].

Random Survival Forests do not make the proportional hazards assumption [23], and have the flexibility to model survival curves of dissimilar shapes for contrasting groups of subjects. Random Survival Forest is an extension of Random Forest, allowing for efficient non-parametric analysis of time–event data [24]. These characteristics allow us to surpass the Cox Regression limitation of the proportional hazard assumption, which requires us to exclude variables that do not fulfill the model assumptions. It has also been shown, by Breiman [24], that ensemble learning can be further improved by injecting randomization into the base learning process (i.e., the use of Random Forests).

Previous researchers have also proposed the integration of several algorithms to improve the performance in the prediction of dropout, such as the use of clustering methods combined with churn prediction [25–27], where the customers are grouped on clusters to improve the prediction performance within each cluster. Clustering methods use unsupervised algorithms to group elements with similar characteristics. Such unsupervised methods have been widely used, employing approaches such Hierarchical Clustering [28], *k*-Means [27], or Random Forest Clustering [24]. Vijaya and Sivasankar [27], following this concept, have suggested the adoption of hybrid models combining more than one classier, in order to increase the performance, compared to the use of single classifiers.

Jafari-Marandi et al. [29] have also explored an approach combining clustering methods in parallel with a classification approach.

Although there have been several studies addressing whether a given customer will dropout or not, there is a lack of research regarding the prediction of when customer will dropout. To address this lack of research, survival analysis can be utilized, which allows for prediction of when the customer will dropout, providing an opportunity to explore the duration of the relation of the customer with the organization and its influence on the churn prediction. Additionally, we assess whether survival analysis combined with the clustering could improve the prediction performance.

In this study, we investigate whether a hybrid approach using clustering and random survival forests, which have never been used to predict membership based on data, can improve the prediction performance compared to a random survival model without the use of clusters. The performance is evaluated by comparing the average discrepancies in the customer status (dropout/non-dropout) in both approaches.

The remainder of this paper is organized as follows: In the next sections, we detail survival analysis, and survival trees. Next, we describe our research methodology. The Section 7 provides the outcomes, addressing our research goal, providing evaluation metrics, and comparing the performance using clusters or without clusters. Finally, the discussion and conclusion are provided.

## 2. Why Dropout Prediction?

The problem related to contractual setting scenarios is that customer dropout is more damaging than in non-contractual settings (e.g., buying a product), representing a well-defined termination of a relationship with a organization [5]. Organizations are more profitable when they retain customers, due to reduced marketing costs and increased sales, minimizing the problems related to reduced sales, competitors gaining new customers, and loss of market [6].

The costs related to getting new customers are five to six times greater, in relation to maintain existing ones [30], which has motivated the organizations to move from capturing customers to retaining them [14].

The overall idea is that investment in retention strategies has higher returns [13]; however, it must be considered that it should be developed in customers with a higher likelihood of churn. It should also be considered that non-profitable retention actions must be avoided [31].

The analysis of existing customer data supports the extraction of patterns related to customer dropout, which an organization can then use to retain customers. Using this information, companies can anticipate churners and develop countermeasures to avoid desertions, retaining as much customers as possible through measures such as giving concessions [27]. By using historical data, organizations can create trained models for the classification of future dropout or non-dropout. This idea allows relevant actors to identify ways to incite customers to stay, by estimating the probability of dropout in a given period of time [15].

## 3. Survival Analysis

Survival analysis focuses on the analysis of the time remaining until an event of interest occurs, and explores its relationships with different factors. The main advantage is related to the concept of censoring, indicating those observations that are not completely related to the event of interest (e.g., customers that have not dropped out yet), which are incorporated into the analysis. This means that there are customers that are still active, for whom we do not know whether the event of dropout has occurred; we call this censorship. Survival models take censoring into account and incorporate this uncertainty—instead of predicting the time of event such in regression models, survival models allow for prediction of probability of an event happening at a particular time.

The time of dropout is represented by T, which is a non-negative random variable, indicating the time period of the event occurring for a randomly selected individual from the population, representing the probability of an event to occurring each time period, given that it has not already occurred in a previous time period; this is known as the discrete-time hazard function [22]. The survival function represents the probability of an individual surviving after time $t$, $S(t) = P(T > t)$, $t \geq 0$, with the properties $S(0) = 1$ and $S(\infty) = 0$. The distribution function is represented by F, defined as $F(t) = P(T \leq 0)$, for $t \geq 0$. The function of probability density represented by $f$ where:

$$f(t) = \lim_{dt \to 0} \frac{P[t \leq T < t + dt]}{dt}, \tag{1}$$

where $f(t)dt$ represents the probability of an event occurring at time $t$. We represent the distribution evolution of the dropout probability with time using the hazard function, represented as:

$$\lambda(t) = \lim_{dt \to 0} \frac{P[t \leq T < t + dt | T \geq t]}{dt}. \tag{2}$$

The determination of the survival curves is based on the following elements: (1) The total value of observations removed during the time period (e.g., days, months, or years), either by dropout or by censorship; (2) observations composing the sample of the study; and (3) customers who have not yet dropped out at any given time. The survival probability until the time period $i$, $(p_i)$ is calculated as:

$$p_i = \frac{r_i - d_i}{r_i}, \tag{3}$$

where $r_i$ is the number of individuals that survived at the beginning of the period and $d_i$ the number of individuals who left during the period. The survival time estimate is also made considering the month in which it is found (estimated). Cox regression allows us to test difference between survival times. The advantage of using survival analysis is that it further allows us to detect whether the risk of an event differs systematically across different people, using specific predictors. The coefficients in the Cox regression are related to the hazard, where a positive value represents a worse prognosis and, in contrast, a negative denotes a better prognosis. The advantage of survival analysis is that it allows us to include information of covariates that were censored up to the censoring event.

The Cox PH model assumes the covariates to be time-independent; for example, gender and age do not change over time when retrieved [32]. As the Cox model requires the hazards in both groups to be proportional, researchers are often asked to "test" whether the hazards are proportional [33]. Considering this, we explored another approach that allowed us to develop the analysis without proportional hazard assumptions; namely, survival trees.

## 4. Survival Trees

Survival trees are methods based on Random Forest models [24]. A Random Survival Forest is an ensemble method for analysis of right-censored data [34], using randomization to improve the performance. Random survival forests follow this framework [34]:

- Draw B random samples of the same size from the original data set with replacement. The samples that are not drawn are said to be out-of-bag (OOB). Grow a survival tree on each of the b = 1 , . . ., B samples.
- At each node, select a random subset of predictor variables and find the best predictor and splitting value that provide two subsets (the daughter nodes) which maximizes the difference in the objective function.
- Repeat step 2 recursively on each daughter node until a stopping criterion is met.
- Calculate a cumulative hazard function (CHF) for each tree and average over all CHF for the B trees to obtain the ensemble CHF.

- Compute the prediction error for the ensemble CHF using only the OOB data.

In each node, a predictor $x$ is selected from a randomly selected predicted variables and split value $c$ (one unique value of $x$). Each sample $i$ if assigned to the daughter right node if $x_i \leq c$, or the left node if $x_i \geq c$. Then, the log-rank is calculated, as follows:

$$L(x,c) = \frac{\sum_{i=1}^{N}\left(d_{i,1} - Y_{i,1}\frac{d_i}{Y_i}\right)}{\sqrt{\sum_{i=1}^{N}\frac{Y_{i,1}}{Y_i}\left(1 - \frac{Y_{i,1}}{Y_i}\right)\left(\frac{Y_i - d_i}{Y_i - 1}\right)d_i}} \tag{4}$$

where

- $j$: Daughter node, $j \in \{1, 2\}$;
- $d_{i,j}$: Number of events at time $t_i$ in daughter node $j$;
- $Y_{i,j}$: Number of elements that had the event or are in risk at time $t_i$ in daughter node $j$;
- $d_i$: Number of events at time $t_i$, such that $d_i = \sum_j d_{i,j}$;
- $Y_i$: Number of elements that experienced an event or are at risk at $t_i$, such that $Y_i = \sum_j Y_{i,j}$.

We loop every $x$ and $c$ until we find $x^*$ that satisfies $|L(x^*, c^*)| \geq |L(x,c)|$ for every $x$ and $c$. The model performance is assessed using the concordance probability (C-index), and Brier Score (BS) [35]. The feature importance is determined by calculating the difference between the true class labels and noised data [24].

The BS is used to evaluate the predicted accuracy of the survival function at a given time $t$. It represents the average square distance between the survival status and the predicted survival probability, where a value of 0 is the best possible outcome.

$$BS(t) = \frac{1}{N}\sum_{i=1}^{N}\left(\frac{\left(0 - \hat{S}(t, \vec{x}_i)\right)^2 \cdot \mathbb{1}_{T_i \leq t, \delta_i = 1}}{\hat{G}(T_i^-)} + \frac{\left(1 - \hat{S}(t, \vec{x}_i)\right)^2 \cdot \mathbb{1}_{T_i > t}}{\hat{G}(t)}\right). \tag{5}$$

The model should have a Brier score below 0.25, considering that, if $\forall i \in [\![1, N]\!]$, $\hat{S}(t, \vec{x}_i) = 0.5$, then $BS(t) = 0.25$.

## 5. Methodology

To achieve our research goals, to simplify the analysis, the survival probabilities are first presented as a survival curve to provide an overall perspective of dropout over time; the representation of the survival probabilities indicates the time where the events are observed [36]. Then, the machine learning survival model was created, following the approach of Ishwaran et al. [34] and using PySurvival [37]. The model performance was determined according to the Brier Score (BS) and Mean Absolute Error (MAE), considering that, due to the censoring of data, standard evaluation metrics such as root mean square error are not suitable [35] in the testing set. The model performance predicted and actual customer dropout are presented as a time-series with the performance indicators Root Mean Square Errors, Median Absolute Error and Mean Absolute Error [37]. The training and testing sets were created using the scikit-learn package with the holdout method (85/25) [38]. The feature importance was determined by calculating the differences between the true class label and noised data [24].

The hybrid model was developed through the identification of an optimal number of clusters. The calculation of the optimal number of clusters was developed based on the Bayesian Information Criterion (BIC), where the model with the lowest score was selected as the best model [39]; however, Scrucca et al. [40] have suggested using the higher BIC score, which we followed. In addition, we used visualization to increase the interpretability of the number of clusters, which was provided using the elbow method. Next, $k$-Means was used to partition the observations into the identified number of clusters.

*5.1. Model Performance Evaluation*

The BS measures the average discrepancy between the status (dropout/non-dropout) and the estimated probabilities at a given time. The Integrated Brier Score (IBS) was used to calculate the performance for all available times (i.e., from $t_1$ to $t_{max}$) as:

$$IBS = \int_{t_1}^{t_{max}} BS^c(t)dw(t). \tag{6}$$

This represents the average square distance between the survival status and the predicted survival probability, where a value of 0 is the best possible outcome.

The Mean Absolute Error (MAE) is a measure of error between observed values and predicted ones, where $y_i$ and $x_i$ are the predicted and the true values, respectively:

$$MAE = \sum_{i=1}^{D} |x_i - y_i|. \tag{7}$$

The IBS and MAE were also calculated for each cluster, in order to conduct a performance comparison against the model without clusters. Additionally, validation tests were performed to compare the accuracy of the hybrid approach against that of the random survival model without clusters. To that end, a paired Mann–Whitney was conducted to estimate whether the prediction ability was significant, with a confidence interval of 95%.

*5.2. Model Operacionalization*

The survival analysis was conducted using the Lifelines package [41] (See Appendix A for software versioning). Dropout was considered a binary value, where one represents churn and zero represents no churn. Dropout happens when a member does not make a payment.

The random survival forest was developed using the PySurvival package [37]. PySurvival is an open-source python package for Survival Analysis modeling. The model was built with 75% of the data for training and 25% for testing.

Using the mclust package [40], the number of clusters was calculated by choosing a varying number of components and identifying the structure of the covariance matrix, based on modelling with a multivariate normal distribution for each component constituting the data set [42].

The hybrid approach was developed as follows:

- Identify the optimal number of clusters using the mclust package of Scrucca et al. [40].
- Fit the model using the identified number of clusters.
- For each element, estimate the cluster membership.
- For each cluster, follow the framework proposed by Ishwaran et al. [34] to calculate the random survival model.

## 6. Data Set

In this study, the data of 5209 Portuguese health club customers were analyzed (mean age = 27.88, SD = 11.80 years). The data were collected using the e@sport software (Cedis, Portugal) between 2014 and 2017. The information retrieved included: Age of the participants (in years), Sex (0, female; 1, male), non-attendance days before dropout, total amount billed, average number of visits per week, total number of visits, weekly contracted accesses, number of registration renewals, number of customer referrals, registration month, customer enrollment duration, and status (dropout/non-dropout). A dropout event is considered to occur when a customer communicates their intention to terminate the contract, or have not paid the monthly fee for 60 days.

Table 1 shows the summary statistics of the data. The average age was 27.9 ± 11.8, and the entries were 29 ± 41.2 with an inscription period of 9 ± 8.2 months. Figure 1 shows the

distribution of the dropout, considering the number of months of membership (where 0 denotes non-dropout and 1 denotes dropout).

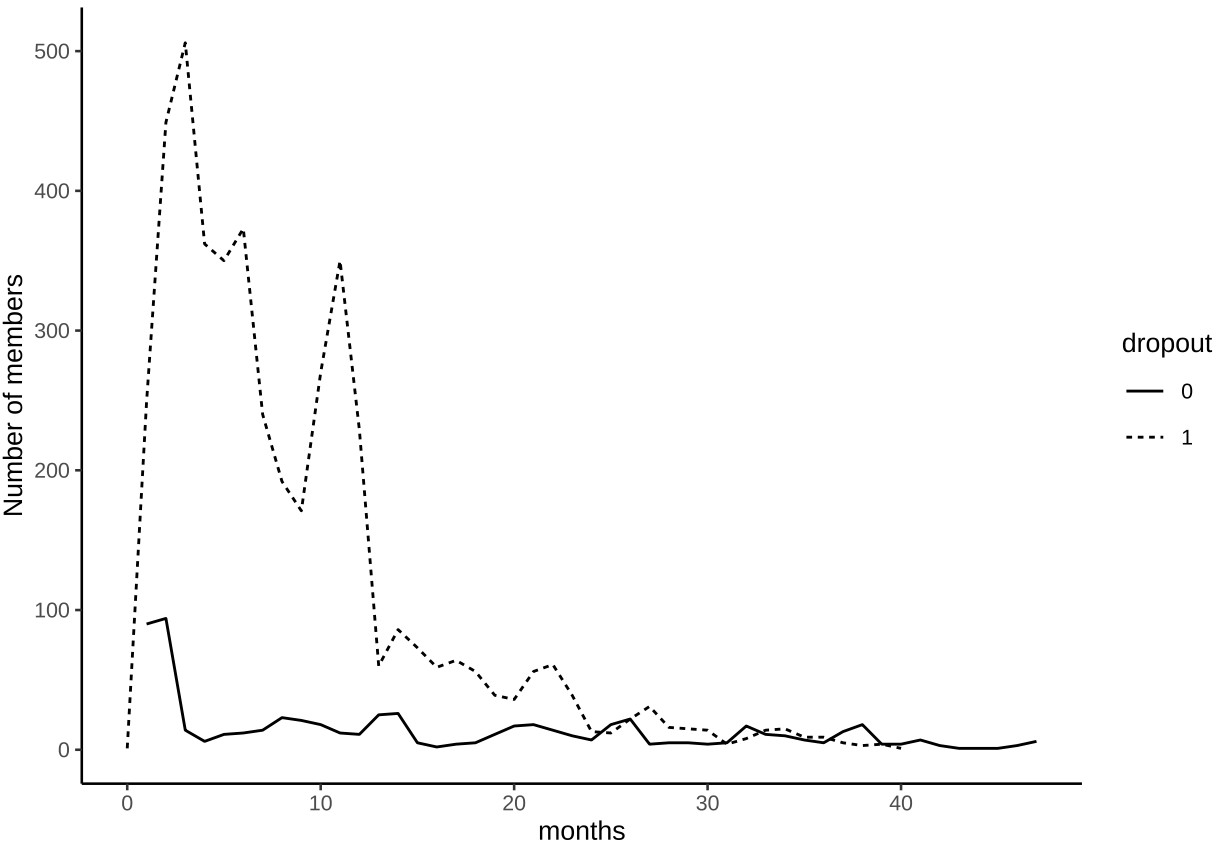

**Figure 1.** Number of members per month.

**Table 1.** Summary statistics of features used.

| Characteristic | N = 5209 |
|---|---|
| Age (age in years), Mean (SD) | 28 (12) |
| Male or female (percentage of male), % | 35% |
| dayswfreq (non-attendance days before dropout), Mean (SD) | 76 (102) |
| tbilled (total amount billed), Mean (SD) | 155 (155) |
| maccess (average entries by week), Mean (SD) | 0.89 (0.76) |
| freeuse (user with free use (1) or with limited entries (0), % | 4.9% |
| nentries (number total of entries), Mean (SD) | 29 (41) |
| cfreq (weekly contracted accesses), % | |
| 2 | 1.3% |
| 4 | 2.4% |
| 6 | 0.2% |
| 7 | 96% |
| months (customer enrolment, in months), Mean (SD) | 9 (8) |
| dropout (customer dropout 1; non-dropout 0), % | 88% |

## 7. Results

Table 2 provides the data regarding the survival time of the customers during the first couple of months. The results indicate that the customers have a survival probability of 24.44% at 12 months (column $p_i$; likelihood probability) with a median survival time of 10 months (column estimated_survival). The survival probability at 6 months was 54.5%, representing a risk of dropout of 45.5% with an estimated survival of 6 months.

**Table 2.** Determination of survival time probabilities.

| Event_at | Removed | Observed | Censored | Entrance | At_Risk | Estimated_Survival | Prob |
|---|---|---|---|---|---|---|---|
| 0 | 1 | 1 | 0 | 5209 | 5209 | 7 | 1.000 |
| 1 | 339 | 249 | 90 | 0 | 5208 | 7 | 0.952 |
| 2 | 543 | 449 | 94 | 0 | 4869 | 7 | 0.864 |
| 3 | 520 | 506 | 14 | 0 | 4326 | 7 | 0.763 |
| 4 | 368 | 362 | 6 | 0 | 3806 | 7 | 0.691 |
| 5 | 361 | 350 | 11 | 0 | 3438 | 6 | 0.620 |
| 6 | 385 | 373 | 12 | 0 | 3077 | 6 | 0.545 |
| 7 | 254 | 240 | 14 | 0 | 2692 | 5 | 0.496 |
| 8 | 215 | 192 | 23 | 0 | 2438 | 6 | 0.457 |
| 9 | 192 | 171 | 21 | 0 | 2223 | 6 | 0.422 |
| 10 | 288 | 270 | 18 | 0 | 2031 | 6 | 0.366 |
| 11 | 362 | 350 | 12 | 0 | 1743 | 9 | 0.293 |
| 12 | 240 | 229 | 11 | 0 | 1381 | 10 | 0.244 |
| 13 | 85 | 60 | 25 | 0 | 1141 | 9 | 0.231 |
| 14 | 112 | 86 | 26 | 0 | 1056 | 9 | 0.212 |
| 15 | 78 | 73 | 5 | 0 | 944 | 9 | 0.196 |
| 16 | 61 | 59 | 2 | 0 | 866 | 10 | 0.183 |
| 17 | 68 | 64 | 4 | 0 | 805 | 10 | 0.168 |
| 18 | 61 | 56 | 5 | 0 | 737 | 9 | 0.155 |
| 19 | 50 | 39 | 11 | 0 | 676 | 9 | 0.146 |
| 20 | 53 | 36 | 17 | 0 | 626 | 9 | 0.138 |
| 21 | 74 | 56 | 18 | 0 | 573 | 10 | 0.124 |
| 22 | 75 | 61 | 14 | 0 | 499 | 11 | 0.109 |
| 23 | 49 | 39 | 10 | 0 | 424 | 11 | 0.099 |
| 24 | 20 | 13 | 7 | 0 | 375 | 10 | 0.096 |

Note: Removed, the sum of customers with dropout and that are censored; Censored, the event did not occur during the period of this data, collection; Risk of Dropout, number of customers at risk of dropout; prob, survival probability; Estimated Survival, months to survive in the health club.

Figure 2 shows the overall Kaplan–Meier survival curve, considering the number of months of membership (*x* axis) against survival probability (*y* axis). The customer dropout is very high in the first 12 months, ranging from a survival probability of 54% after the first 6 months to 24.44% after 12 months.

The survival considering other cohorts is represented in Figure 3; in particular survival by gender and survival by contracted frequency. The survival by gender was very similar; however, that by contracted access frequency indicated that customers with a contracted access frequency of 6 and 4 times a week have higher survival probabilities, compared to the lower survival of customers with contracted access frequencies of 7 and 2 times a week. Survival curves can explore tendencies related to survival, in order to extract actionable knowledge, giving a perspective regarding the probability to survival within a given period of time.

The proportional hazard assumptions failed in the following variables: Age ($p < 0.01$), cfreq ($p < 0.01$), dayswfreq ($p < 0.01$), tbilled ($p < 0.01$), freeuse ($p < 0.01$), and nentries ($p < 0.01$). Therefore, it was not possible to calculate the effect of the cohorts, in terms of the survival time, using the Cox regression.

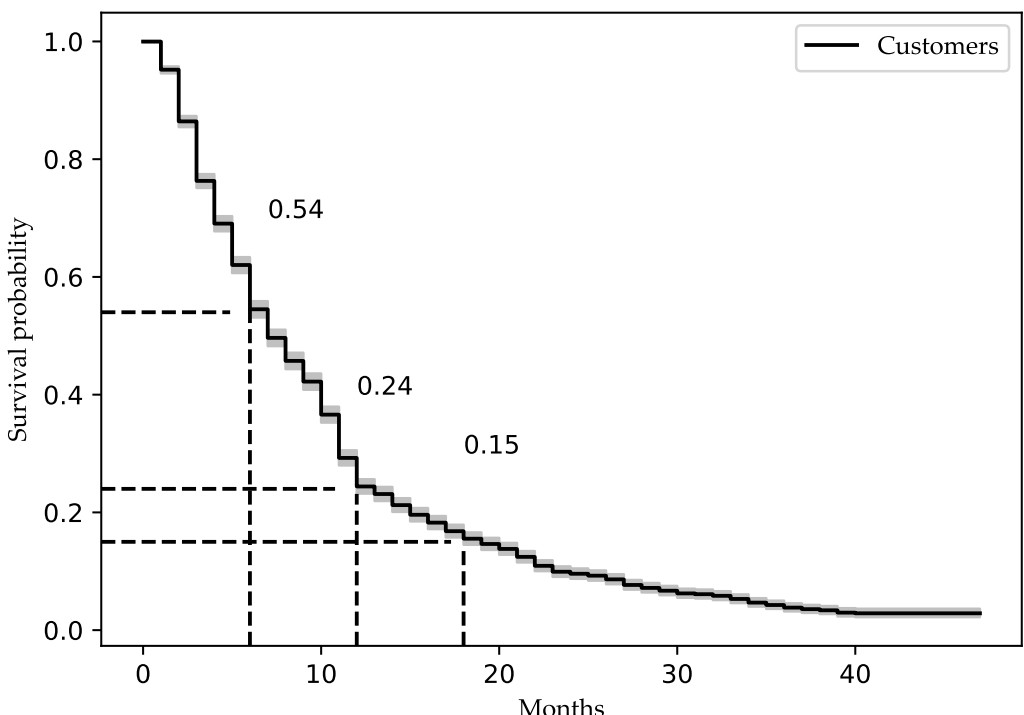

**Figure 2.** Survival probabilities.

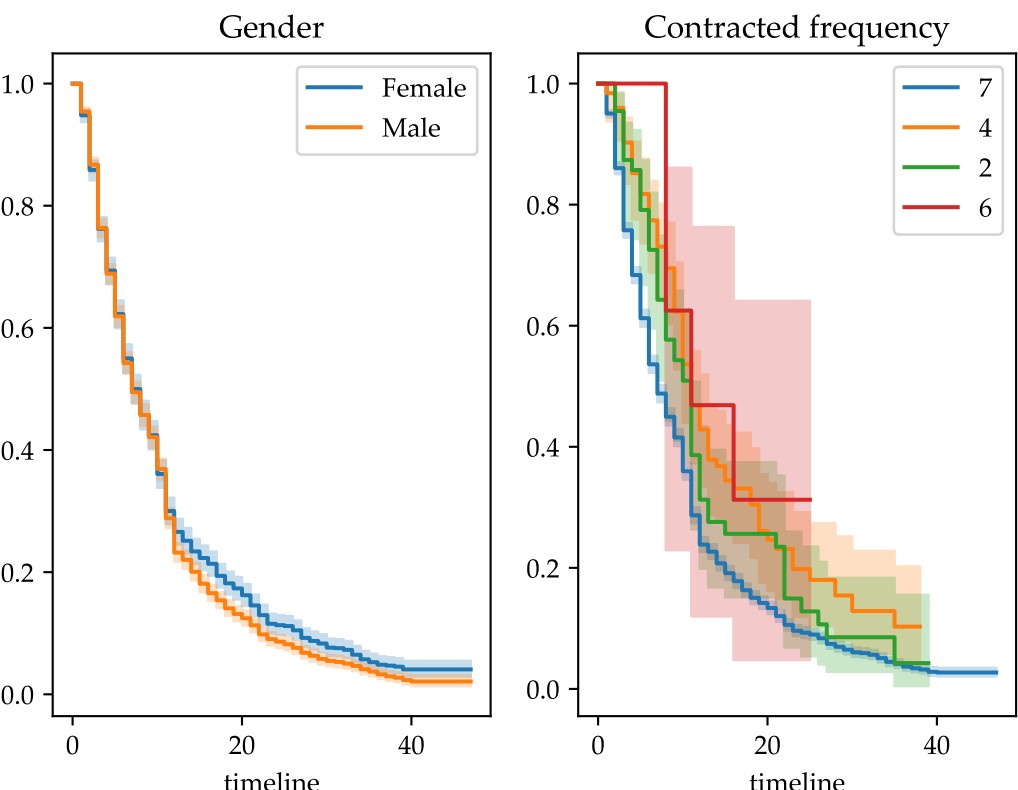

**Figure 3.** Survival by gender and contracted access frequency.

### 7.1. Survival Trees

To evaluate the performance of the random survival forest in predicting the survival time, considering the effect of the cohorts, we calculated the concordance probability (C-

index), IBS, and Mean Absolute Error (MAE). The IBS presented an accuracy along the 12 months of 0.08 (Figure 4).

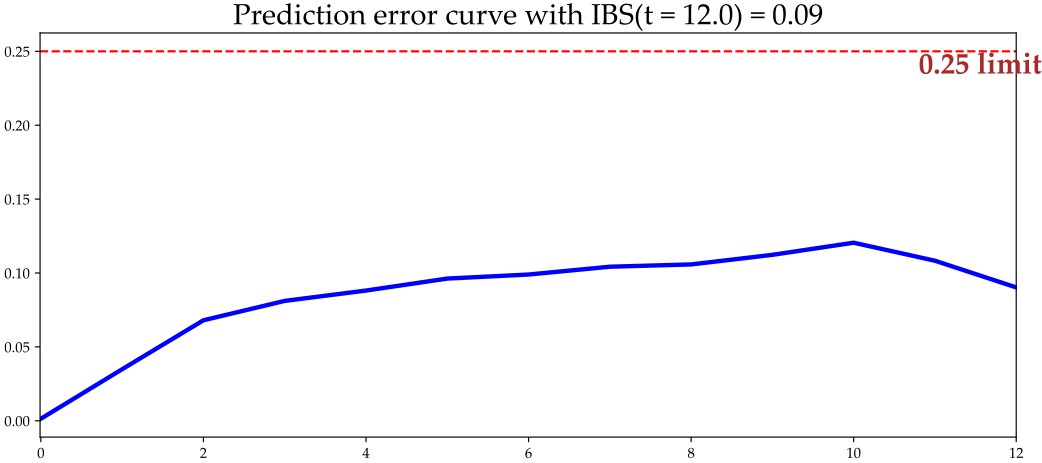

**Figure 4.** Global Model performance.

Figure 5 presents the actual and predicted customers that dropped out during the 40 months, showing an average absolute error of 7.2 customers.

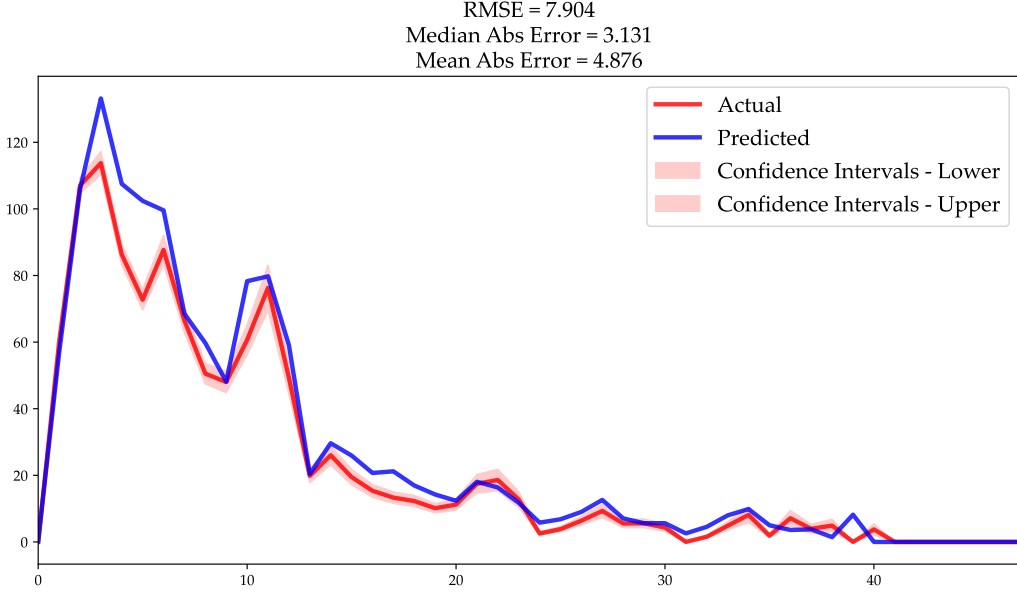

**Figure 5.** Global model performance predicted versus actual.

Table 3 shows the feature importance, calculated according to Breiman [24], with the percent increase in misclassification rate, as compared to the out-of-bag rate (with all variables intact). Out-of-bag is a bootstrap aggregating method (i.e., sub-sampling with replacement to create training samples for the model to learn from), where two independent sets are created. One set, the bootstrap sample (in-the-bag), is obtained by sampling with replacement from the from the original data set; meanwhile, the out-of-bag is the difference between the original data set and the bootstrap data set. The most important variable was *tbilled*, followed by *dayswfreq* and *nentries*. The least relevant features where *cfreq*, *age*, and *sex*.

**Table 3.** Feature importance in the survival model.

| Feature | Importance | Pct_Importance |
|---|---|---|
| tbilled | 7.639 | 0.264 |
| freeuse | 5.758 | 0.199 |
| dayswfreq | 5.105 | 0.176 |
| nentries | 4.847 | 0.167 |
| maccess | 3.744 | 0.129 |
| cfreq | 1.871 | 0.065 |
| sex_1 | −0.057 | 0.000 |
| age | −0.210 | 0.000 |

The prediction was very similar to the actual value. The model accuracy was very high, with a root mean square error of 8. The mean absolute error mean was 4.88 customers, and the median absolute error was 3.13 customers.

### 7.2. Survival Tree-Based Model with Clusters

In our approach, we have created clusters and applied the survival trees within each cluster. The determination of the clusters using the BIC criterion and the EEV model was as follows: 9 clusters, 6990.94; 7 clusters, −30,105.59; and 6 clusters, −44,616.29. Figure 6 shows the determination of the number of clusters using the BIC. The elbow analysis, presented in Figure 7, shows that the curve flattened after 8 clusters. Therefore, nine clusters was considered the optimal number of clusters, which was used to partition the customers. The higher prediction performance is represented in the three bigger clusters, considering the number of elements cluster 0 ($n = 1955$), cluster 4 ($n = 729$), and cluster 8 ($n = 1020$).

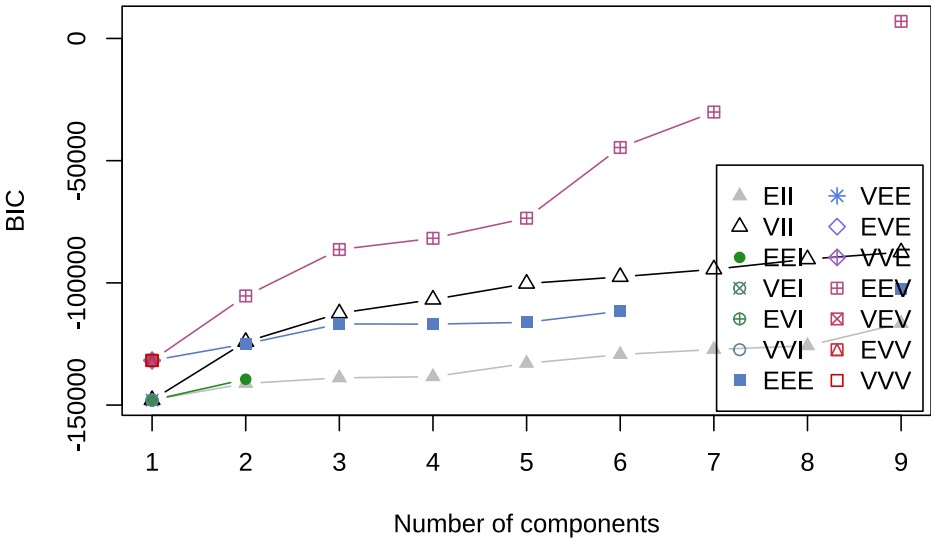

**Figure 6.** Cluster number analysis.

The prediction performance analysis (IBS score) in clusters 0, 4, and 9, yielded accuracies of 0.07 (Figure 8), 0.078 (Figure 9) , and 0.105 (Figure 10), respectively. The cluster 0 actual versus predicted model had a mean absolute error of 4.9 customers, median absolute error of 0.864, and the Root Mean Square Error of 8.96 (Figure 11). The feature importance in the survival model with cluster 0 (Table 4) identified the three most relevant features to predict survival as *freeuse*, *age*, and *maccess*, while the features with lower relevance were *nentries*, *dayswfreq*, and *tbilled*.

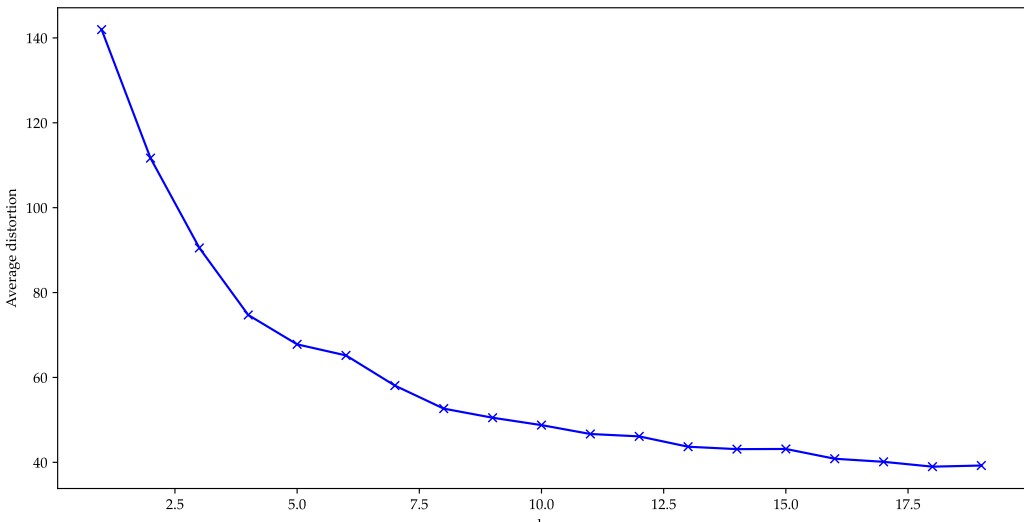

**Figure 7.** Elbow analysis.

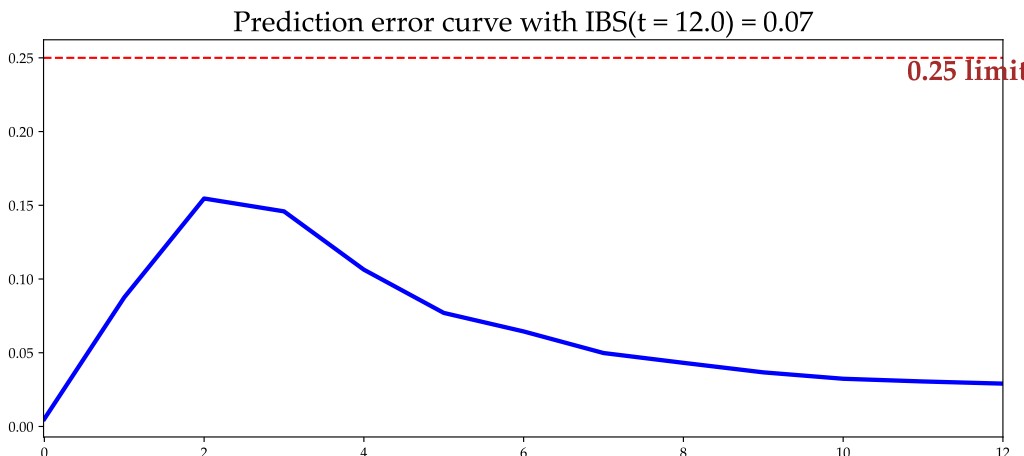

**Figure 8.** Model performance for cluster 0.

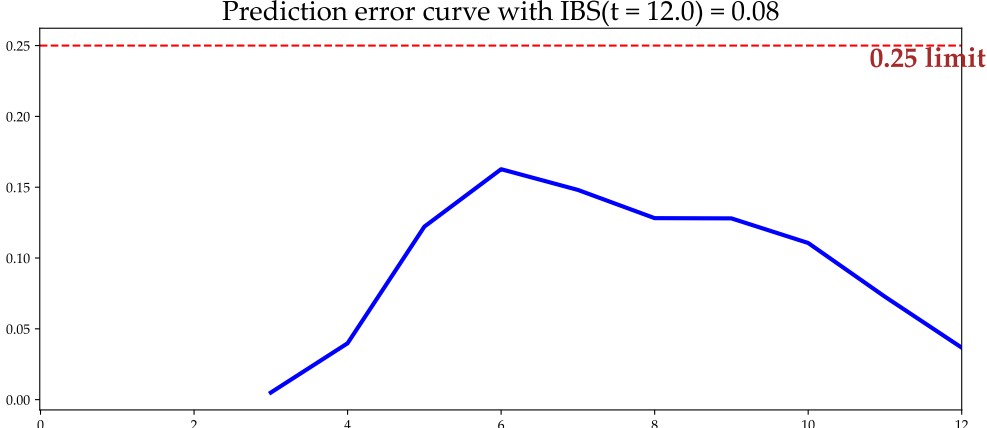

**Figure 9.** Model performance for cluster 4.

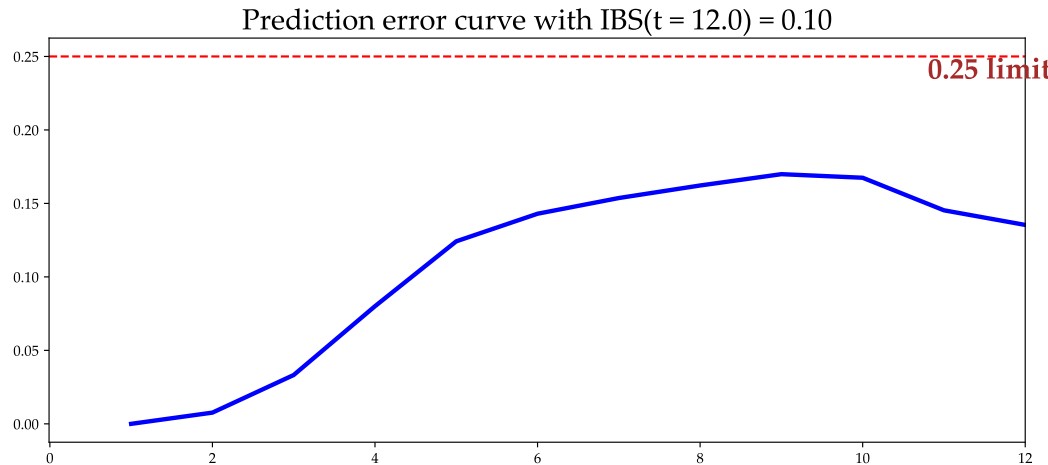

**Figure 10.** Model performance for cluster 8.

**Table 4.** Feature importance in the survival model with cluster 0.

| Feature | Importance | Pct_Importance |
|---|---|---|
| freeuse | 1.586 | 0.443 |
| age | 1.052 | 0.294 |
| maccess | 0.944 | 0.263 |
| sex_1 | −0.033 | 0.000 |
| cfreq | −1.026 | 0.000 |
| nentries | −1.595 | 0.000 |
| dayswfreq | −1.960 | 0.000 |
| tbilled | −3.245 | 0.000 |

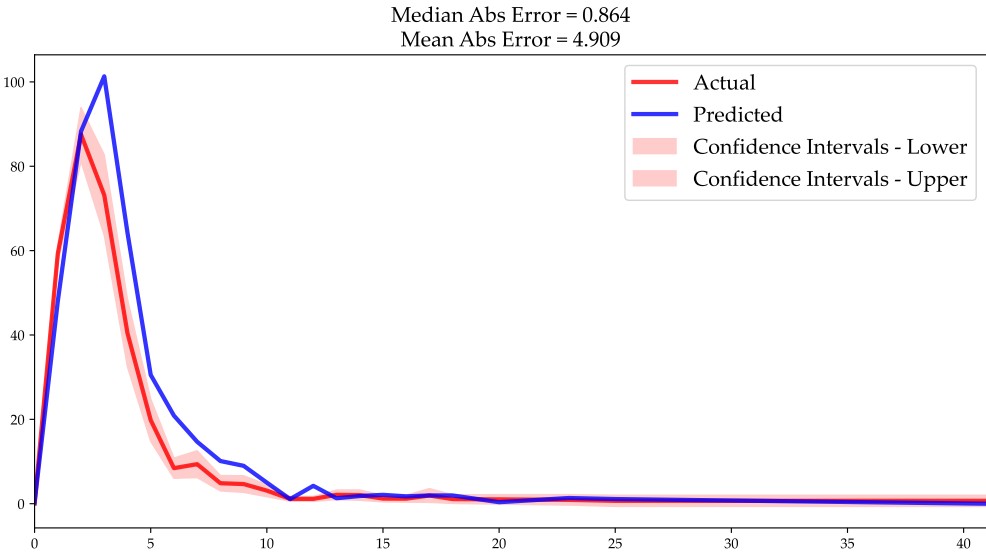

**Figure 11.** Conditional survival forest for cluster 0.

The cluster 4 actual versus predicted model presented a mean absolute error of 2.29 customers, median absolute error of 0.717, and Root Mean Square Error of 3.32 (Figure 12). The feature importance in the survival model with cluster 4 (Table 5) identified the three most relevant features to predict survival as *nentries*, *dayswfreq*, and *tbilled*, while the least relevant were *freeuse*, *cfreq*, and *sex*.

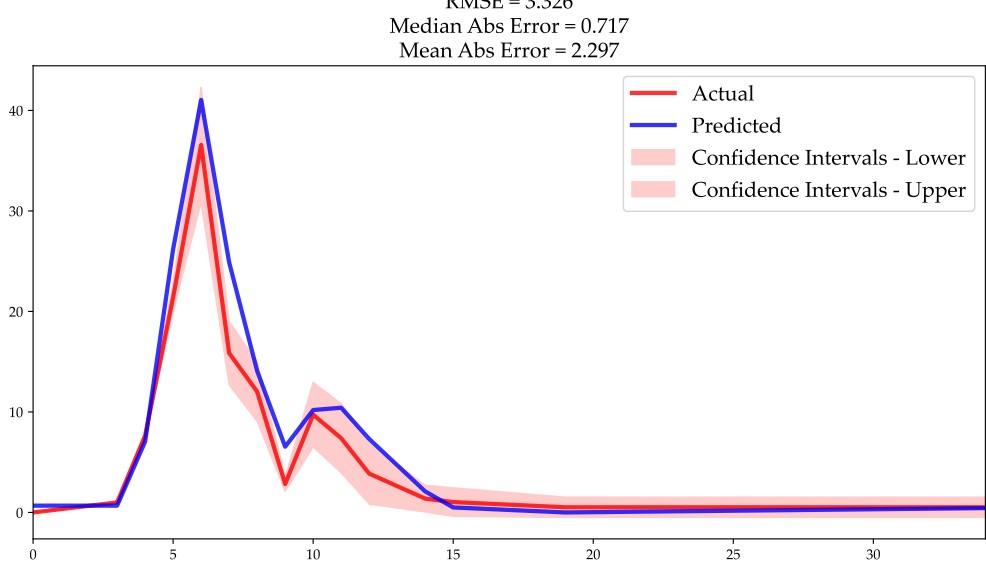

**Figure 12.** Conditional survival forest for cluster 4.

**Table 5.** Feature importance in the survival model with cluster 4.

| Feature | Importance | Pct_Importance |
|---------|------------|----------------|
| nentries | 2.226 | 0.388 |
| dayswfreq | 2.086 | 0.363 |
| tbilled | 1.033 | 0.180 |
| age | 0.225 | 0.039 |
| maccess | 0.174 | 0.030 |
| freeuse | 0.000 | 0.000 |
| cfreq | 0.000 | 0.000 |
| sex_1 | −0.989 | 0.000 |

Finally, cluster 8 actual versus predicted model presented a mean absolute error of 2.02 customers, median absolute error was 1.09, and Root Mean Square Error of 3.52 (Figure 13). The feature importance in the survival model with cluster 8 (Table 6) identified the three most relevant features to predict survival as *maccess*, *dayswfreq*, and *tbilled*, while the least relevant were *cfreq*, *age*, and *sex*.

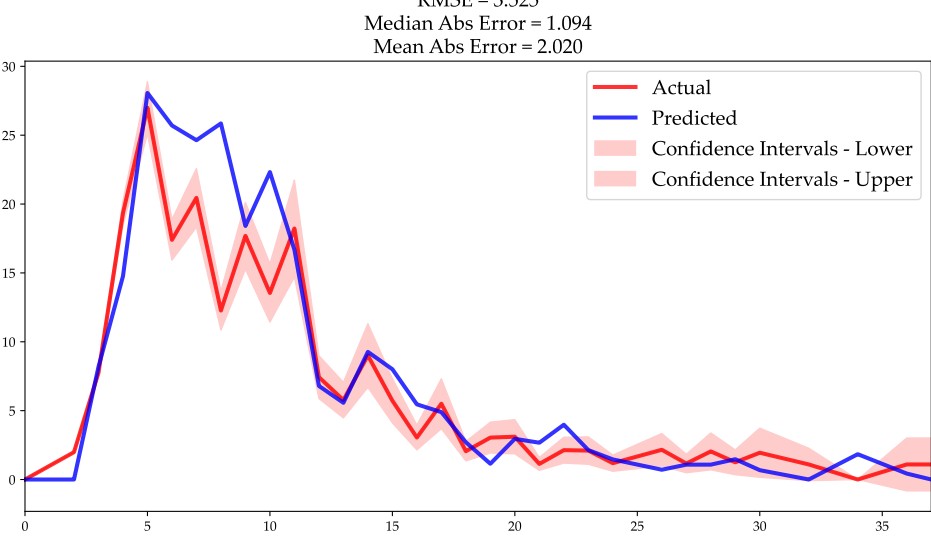

**Figure 13.** Conditional survival forest for cluster 8.

**Table 6.** Feature importance in the survival model with cluster 8.

| Feature | Importance | Pct_Importance |
|---|---|---|
| maccess | 4.632 | 0.278 |
| dayswfreq | 3.235 | 0.194 |
| tbilled | 2.962 | 0.178 |
| nentries | 2.366 | 0.142 |
| freeuse | 2.191 | 0.132 |
| cfreq | 1.026 | 0.062 |
| age | 0.237 | 0.014 |
| sex_1 | −0.626 | 0.000 |

Model Comparison

Table 7 shows the performance of both approaches; that is, with and without clusters. The RMSE, mean, and median in the clustered approach was lower than that when not using clusters to predict the survival time until dropout. The metrics, comparing the actual values in each cluster against the model without clusters, were as follows: median $1.68 \pm 1.34$ vs. 4.8 w/o clusters; mean $1.348 \pm 0.831$ vs. 3.13 w/o clusters, RMSE $2.80 \pm 2.69$ vs. 7.9 w/o clusters, and IBS $0.051 \pm 0.036$ vs. 0.089 w/o clusters. On average, the results outperform the model without clusters even if we consider plus one standard deviation in all the indicators.

**Table 7.** Brier Score performance prediction in each cluster.

| Cluster | RMSE | Mean | Median | IBS | $n$ | Ntrain | Ntest |
|---|---|---|---|---|---|---|---|
| 0 | 9.181 | 0.893 | 4.766 | 0.070 | 1955 | 1466 | 489 |
| 1 | 1.350 | 0.938 | 1.065 | 0.000 | 109 | 81 | 28 |
| 2 | 3.861 | 1.159 | 2.272 | 0.036 | 425 | 318 | 107 |
| 3 | 2.156 | 1.207 | 1.529 | 0.057 | 624 | 468 | 156 |
| 4 | 2.903 | 0.882 | 1.891 | 0.078 | 729 | 546 | 183 |
| 5 | 0.312 | 0.000 | 0.127 | NaN | 49 | 36 | 13 |
| 6 | 0.689 | 0.400 | 0.521 | NaN | 40 | 30 | 10 |
| 7 | 1.255 | 1.025 | 1.050 | 0.017 | 258 | 193 | 65 |
| 8 | 3.513 | 0.975 | 1.941 | 0.105 | 1020 | 765 | 255 |
| w/cluster | 7.904 | 3.131 | 4.876 | 0.089 | 5209 | 3906 | 1303 |

Note: NaN value not possible to calculate; $n$ represents the number of elements in each cluster; Ntrain and Ntest are the number of elements used to train and test the model, respectively.

Overall, the performance was improved through clustering. The performance is also better using mean and median.

The model without clusters presented a RMSE of 7.904, the mean absolute error mean was customers, the median absolute error was 4.876, and the IBS was 0.089. One cluster in the model using clusters had worse performance (cluster 0), with a RMSE 9.181, mean absolute error of 0.893, and median absolute error 4.766. The cluster with the best performance (cluster 5) had a RMSE of 0.312, mean absolute error of 0, and median absolute error 0.127. The overall performance of the model was improved when using clusters, with the non-clustering IBS of 0.089 only being surpassed by cluster 8, with an IBS of 0.105.

Comparing the prediction accuracy in each cluster using the Brier Score, the median Brier score without clusters was 0.356, while those for clusters 0, 1, 2, 3, 4, 5, 6, 7, and 8, were 0.022, 0.114, 0.041, 0.056, 0.022, 0.072, 0.133, 0.137, and 0.049, respectively. The results of applying the Mann–Whitney test between clusters and without clusters were as follows: cluster 0 (U = 530, $n1 = 19$, $n2 = 29$, $p < 0.05$); cluster 1 (U = 400, $n1 = 19$, $n2 = 25$, $p < 0.05$); cluster 2 (U = 374, $n1 = 19$, $n2 = 21$, $p < 0.05$); cluster 3 (U = 721, $n1 = 19$, $n2 = 42$, $p < 0.05$); cluster 4 (U = 415, $n1 = 19$, $n2 = 23$, $p < 0.05$); cluster 5 (U = 220, $n1 = 19$, $n2 = 13$, $p < 0.05$); cluster 6 (U = 276, $n1 = 19$, $n2 = 19$, $p < 0.05$); cluster 7 (U = 597, $n1 = 19$, $n2 = 39$, $p < 0.05$); and cluster 8 (U = 663, $n1 = 19$, $n2 = 38$, $p < 0.05$). The prediction was statistically significant,

when comparing the prediction accuracy of the survival model using clusters against that without clusters; namely, the median value was lower using clusters.

## 8. Discussion

In this study, we evaluated the performance of random survival forests using membership data of customers of a health club. A survival model was created to determine the duration of the relationship. This approach provides an additional view to identify when the customer dropout will occur, allowing for the development of retention strategies, considering the timings of the events. More than 70% of the customers were predicted to dropout in the first 12 months, which is very high, and has not been identified in other studies. Burez and Vandenpoel [19], in a study of pay TV users, have found that one out of three customers leave the company before one year, and half the customers leave within two years.

The accuracy calculated using the actual and predicted customers who dropped out during the 40 months showed a mean absolute error of 7.5 customers. Using the hybrid model, the mean absolute errors were 1.56, 6.52, and 1.56 customers in clusters 1,2, and 3, respectively. The features *dayswfreq*, *tbilled* and *nentries* represented more than 66% of the importance predicting the survival model without clusters. Accordingly, in the hybrid approach, the most relevant features were *nentries*, *tbilled*, and *dayswfreq*, representing 67% of the importance in the cluster 1; in Cluster 2, *dayswfreq*, *tbilled*, and *nentries* represented 70% in prediction importance; and, in Cluster 3, *tbilled*, *dayswfreq*, and *nentries* representing 69% in the prediction importance.

The use of clustering in supporting customer segmentation to improve the performance of machine learning techniques is not new. Jafari-Marandi et al. [29] have also explored the use of clusters to improve the prediction accuracy. However, this approach combined with the use of survival models, to the best of our knowledge, has not been previously attempted or reported.

The better performance of the hybrid model in predicting when customers will dropout, using existing data, supports the development of management counter-measures to reduce dropout. The duration of the relationship between the customer and the organization is an important aspect, allowing us to understand that the decision of the customer to dropout changes over time, which implies that existing models predicting customer dropout may be only correct at a specific point in time, after which their decision may change [11].

The time perspective allows us to identify the period in which retention actions should be developed; therefore, the prediction should be as accurate as possible. However, customers who are about to churn but cannot be retained should be excluded from the countermeasures to avoid dropout, considering that targeting them may constitute a waste of scarce resources [15]. Addressing only the performance of predictive models considering only accuracy seems to be a reduced perspective, considering that customers with a higher risk of churning may not be the best targets for the development of retention strategies. Further research should be conducted exploring this perspective, thus providing further insight into the return on investment in the development of countermeasures. A business context, or the clarification of business objectives underlying the prediction of customer dropout, should be developed, in order to clarify which objectives should be achieved before the employment of machine learning algorithms.

## 9. Conclusions

In this paper, we investigated the customer dropout in a Health Club organization, considering the dynamic perspective that the dropout risk varies with time. We explored two approaches: using a survival model based on random forests with or without clusters. The model using clusters allowed us to combine the customers into different clusters, comprising a hybrid approach. Based on the results, the performance of the proposed model using clusters presented improved accuracy of the survival model, allowing for the



development of targeted approaches taking into account the timing of when the dropout occurs, considering the cluster a given customer belongs to. Most importantly, managers can use the resulting information for improvement of their retention strategies.

**Author Contributions:** Conceptualization, P.S., J.B., D.M. and J.G.-A.; methodology, P.S. and J.B.; software, P.S. and J.B.; validation, J.B.; formal analysis, P.S. and J.B.; investigation, P.S., J.B. and J.G.-A.; data curation, P.S.; writing—original draft preparation, P.S. and J.B.; writing—review and editing, P.S., J.B. and D.M.; supervision, J.B., D.M. and J.G.-A.; project administration, J.B. and J.G.-A. All authors have read and agreed to the published version of the manuscript.

**Funding:** This research was funded by Spanish Ministry of Science and Innovation under Project PID2021-124054OB-C31, in part by the 4IE+ Project by the Interreg V-A España-Portugal (POCTEP) (2014–2020) Program under Grant 0499-4IE-PLUS-4-E, in part by the Department of Economy, Regional Ministry of Economy, Science and Digital Agenda under Grant GR21133.

**Data Availability Statement:** Not applicable.

**Conflicts of Interest:** The authors declare no conflict of interest. The founding sponsors had no role in the design of the study; in the collection, analyses, or interpretation of data; in the writing of the manuscript, an in the decision to publish the results.

## Appendix A. Software Versioning

```
R version 4.2.1 (2022-06-23)
Platform: x86_64-pc-linux-gnu (64-bit)
Running under: Ubuntu 20.04.4 LTS

Matrix products: default
BLAS:   /usr/lib/x86_64-linux-gnu/openblas-pthread/libblas.so.3
LAPACK: /home/sobreiro/miniconda3/envs/survival/lib/libmkl_intel_lp64.so

locale:
[1] en_US.UTF-8

attached base packages:
[1] stats     graphics  grDevices utils     datasets  methods   base

other attached packages:
 [1] mclust_5.4.10    labelled_2.9.1   kableExtra_1.3.4 gtsummary_1.6.0
 [5] visdat_0.5.3     readxl_1.4.0     stargazer_5.2.3  reticulate_1.25
 [9] ggplot2_3.3.6    dlookr_0.5.6     dplyr_1.0.9

loaded via a namespace (and not attached):
 [1] reactable_0.2.3     webshot_0.5.3       httr_1.4.3
 [4] tools_4.2.1         utf8_1.2.2          R6_2.5.1
 [7] rpart_4.1.16        DBI_1.1.3           colorspace_2.0-3
[10] withr_2.5.0         tidyselect_1.1.2    gridExtra_2.3
[13] curl_4.3.2          compiler_4.2.1      extrafontdb_1.0
[16] cli_3.3.0           rvest_1.0.2         gt_0.5.0
[19] xml2_1.3.3          labeling_0.4.2      bookdown_0.26
[22] scales_1.2.0        mvtnorm_1.1-3       rappdirs_0.3.3
[25] systemfonts_1.0.4   stringr_1.4.0       digest_0.6.29
[28] rmarkdown_2.14      svglite_2.1.0       pkgconfig_2.0.3
[31] htmltools_0.5.2     showtext_0.9-5      extrafont_0.18
[34] fastmap_1.1.0       highr_0.9           htmlwidgets_1.5.4
[37] rlang_1.0.2         rstudioapi_0.13     sysfonts_0.8.8
[40] shiny_1.7.1         generics_0.1.2      farver_2.1.0
[43] jsonlite_1.8.0      magrittr_2.0.3      Formula_1.2-4
```

```
[46] Matrix_1.4-1       Rcpp_1.0.8.3       munsell_0.5.0
[49] fansi_1.0.3        gdtools_0.2.4      partykit_1.2-15
[52] lifecycle_1.0.1    stringi_1.7.6      yaml_2.3.5
[55] inum_1.0-4         grid_4.2.1         hrbrthemes_0.8.0
[58] promises_1.2.0.1   forcats_0.5.1      crayon_1.5.1
[61] lattice_0.20-45    haven_2.5.0        splines_4.2.1
[64] hms_1.1.1          knitr_1.39         pillar_1.7.0
[67] glue_1.6.2         evaluate_0.15      pagedown_0.18
[70] broom.helpers_1.7.0 vctrs_0.4.1       png_0.1-7
[73] httpuv_1.6.5       Rttf2pt1_1.3.10    cellranger_1.1.0
[76] gtable_0.3.0       purrr_0.3.4        tidyr_1.2.0
[79] assertthat_0.2.1   xfun_0.31          mime_0.12
[82] libcoin_1.0-9      xtable_1.8-4       later_1.3.0
[85] survival_3.3-1     viridisLite_0.4.0  tibble_3.1.7
[88] showtextdb_3.0     ellipsis_0.3.2
```

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
