# Peer review of "Hybrid Random Forest Survival Model to Predict Customer Membership Dropout"

_electronics, doi:10.3390/electronics11203328_

Round 1
Reviewer 1 Report
Add some explanation in Abstract
New references add 2-5 should be related your work.
In introduction part need to add some paragraphs
Author Response
We would like to thank to the reviewer for his suggestions.
Regarding to the issues:
Abstract:
Thanks for pointing this out. We added more explanation in the abstract, following the suggestion of a reviewer to place a statistical testing comparing the two approaches. We hope that the changes are attuned with the reviewer expectations.
Add 2-5 five references:
Thanks for the suggestion. The changes are inserted in the line 47 and signaled at bold.
Introduction:
Was added more text content to the introduction. Hopefully this meet the reviewer expectations, which is highlighted at bold starting at line 29.
Reviewer 2 Report
This study aimed to investigate whether a hybrid approach using clusters and random survival forests improves predictive performance against a random forest model without using clusters in the prediction of customer membership dropout. The experiment did not show significant improvement by the hybrid approach.
Major comments:
1. Statistical testing should be performed to compare the predictive performance between these two approaches.
2. The predictive performance of the model in the whole testing set should be presented.
3. It is difficulty for readers to interpret Table 1. Each variable should be defined clearly.
4. The samples were classified into 9 clusters. The number of samples in each cluster should be provided for both the training and testing sets.
5. Some passages are redundant. The writing needs to be improved. It is difficult for reviewers to evaluate the current form of the manuscript.
Author Response
The submission has been updated and improved based on the comments and suggestions received. Many of the issues raised in the reviews had been discussed by the authors when preparing the original proposal, but we see now that they were not adequately described in the submission. We want to thank to the reviewer for their work on the paper, which has helped us to prepare a better version of our proposal.
Statistical testing should be performed to compare the predictive performance between these two approaches
Thanks for this suggestion. Was added statistical testing to compare the performance of the two approaches.
It is difficulty for readers to interpret Table 1. Each variable should be defined clearly
Once again, thanks for pointing this out, we have added the description of the variables to the table 1.
The samples were classified into 9 clusters. The number of samples in each cluster should be provided for both the training and testing sets
Thanks for reviewing this also, I have added ntrain and nsets to table 7, to complementing existing information as correctly suggested.
Some passages are redundant. The writing needs to be improved. It is difficult for reviewers to evaluate the current form of the manuscript.
To address this issue we improved the manuscript writing, hopefully meets the reviewer expectations. Several results presenting the performance of cluster 0,4 and 8 where merged. Other sentences where integrated, e.g. explanation in the methodology repeated in the results. Once again thanks for pointing this out.

Reviewer 3 Report
The paper presents a novel machine learning approach to eletronic problems. Overall, the manuscript is well written. I'd recommend minor revision as I think the English may need some polishing.
Some minor comments:
- Whats the meaning of the features in Table 3 and 4?
- What do the author mean by "in-the-bag"?
- How does the method compares to the performance of gradient boosting?
Author Response
The submission has been updated and improved based on the comments and suggestions received. Many of the issues raised in the reviews had been discussed by the authors when preparing the original proposal, but we see now that they were not adequately described in the submission. We want to thank the reviewer for their work on the paper, which has helped us to prepare a better version of our proposal.
Whats the meaning of the features in Table 3 and 4?
Thanks for pointing this out. Was added a description of the features to table 1
What do the author mean by “in-the-bag”?
The original text was: "One set, the bootstrap sample, data chosen to be “in-the-bag” by sampling with replacement and the out-of-bag is all data not chosen in the sampling process."
Which was replaced by "One set, the bootstrap sample (in-the-bag) by sampling with replacement created from the original dataset and the out-of-bag, which is the difference between the original dataset and bootstrap dataset."
We hope that this meets the reviewer's expectations.
How does the method compares to the performance of gradient boosting?
Thanks for the suggestion, unfortunately, the package that we used doesn’t employ gradient boosting, only Survival Support Vector Machines.
Round 2
Reviewer 2 Report
Major comments:
1. In addition to predictive performance for each cluster, the overall predictive performance of the hybrid model in the whole testing set should be presented and compared to the model without clustering.
2. As indicated in my previous report, the writing needs to be improved. Extensive editing of English language and style is suggested.
3. As the authors indicated that root mean square error is not suitable, why did they state, "The model accuracy is very high with a root mean square error of 8"?